



# Effects of moss restoration on soil erosion and soil water content in a temperate vineyard

Corinna Gall[1], Silvana Oldenburg[1], Martin Nebel[2], Thomas Scholten[1], Steffen Seitz[1]

[1] Soil Science and Geomorphology, Department of Geosciences, University of Tübingen, Rümelinstr. 19–23, 72070 Tübingen, Germany
[2] Nees Institute for Biodiversity of Plants, University of Bonn, Meckenheimer Allee 170, 53115 Bonn, Germany

*Correspondence to*: Corinna Gall (corinna.gall@uni-tuebingen.de)

**Abstract.**

Soil erosion is a serious problem worldwide, as it jeopardizes soil fertility and thus food security. At the same time, agriculture itself is one of the biggest drivers of soil erosion, and vineyards in particular are vulnerable due to often steep slopes, fragile soils, and management practices. Therefore, the search for alternative management practices becomes vital. Since soil erosion is reduced by vegetation cover, this also applies to moss cover. However, research on the restoration and protection of bare soil using mosses as erosion control is still in its infancy. In this study, the restoration of mosses was investigated by applying artificially cultivated moss mats in a temperate vineyard. The effects of moss restoration on surface runoff and sediment discharge were examined compared to bare soil and cover crops using rainfall simulations at three measurement times during one year (April, June, and October). Additionally, soil water content was monitored for each treatment during all rainfall simulations.

Mosses initially showed considerable desiccation in summer, whereupon their growth declined. In October, the mosses recovered and re-established themselves in the vineyard, showing a high level of resistance. Moss restoration significantly reduced surface runoff by 71.4% and sediment discharge by 75.8% compared to bare soils. While moss restoration had a slightly better effect on reducing runoff and a slightly lower effect on reducing erosion than cover crops (68.1% and 87.7%, respectively), these differences were not statistically significant. Sediment discharge varied seasonally for moss restoration, especially from April to June, which is most likely due to the decline in moss cover and the foliage of the vines in June, as concentrated canopy drip points have formed on the leaves and woody surfaces of the vines, increasing erosion. In April and June, the different treatments do not significantly impact soil water content, while in October, bare soil had the highest and moss restoration the lowest soil water content. According to this, the influence of soil cover varies seasonally, with moss restoration not having a detrimental effect on the soil water content in the drier summer months, but retaining the least water in October. Overall, moss restoration proved to be an appropriate and low-maintenance alternative for erosion control, as it requires no mowing and does not reduce near-surface soil water content during summer.



# 1 Introduction

Soil erosion poses a serious threat to global soil fertility and, consequently, to food security (Amundson et al., 2015). As one of the primary drivers of this issue, agricultural activities exacerbate soil degradation (Borrelli et al., 2017), thus resulting in soils that are no longer able to provide important ecosystem services such as filtering and storing water, providing nutrients, storing carbon, providing habitat for biological activity, and producing biomass (Vogel et al., 2019; FAO and ITPS, 2015). With progression of land use changes and climate change, soil erosion will intensify in the future, which requires the rapid development of effective soil conservation strategies (Olsson et al., 2019; Borrelli et al., 2020).

Vineyards are particularly susceptible to soil erosion due to their typically steep slopes, fragile soils characterized by an extremely basic or acidic pH, loamy or clayey textures, and low soil organic carbon (SOC) contents as well as specific management practices such as fallow interlines (Biddoccu et al., 2017; Rodrigo-Comino, 2018). For instance, conventional farming in vineyards usually involves practices to control weed by means of application of herbicides and tillage that leave the soil bare (Biddoccu et al., 2016), which is the most relevant anthropogenic factor for increased soil erosion in viticulture (Rodrigo-Comino, 2018). As a result of soil erosion in viticulture, the grape yield can decrease by up to 50% according to Costantini et al. (2018), who conducted a multidisciplinary study in nineteen European and Turkish vineyards. In addition, this study emphasizes that soil erosion has degraded essential parameters of soil fertility such as available water capacity, chemical fertility, total nitrogen, and cation exchange capacity, among others. Given the critical role of vineyards in agriculture and their vulnerability to erosion, it is imperative to explore alternative management practices that can mitigate soil erosion effectively. Vegetation cover is well-documented as a natural barrier against soil erosion due to its ability to stabilize the soil and reduce surface runoff (Morgan, 2005). In viticulture, organic management practices that cover the soil surface with vegetation are regularly used, which has been proven in several studies to substantially reduce surface runoff and soil erosion (Seeger et al., 2019; Kirchhoff et al., 2017; Biddoccu et al., 2017; Bagagiolo et al., 2018). These practices include allowing spontaneous vegetation to grow, seeding grasses and cover crops (Morvan et al., 2014; Kirchhoff et al., 2017), applying mulching techniques (Prosdocimi et al., 2016), or planting aromatic herbs (Dittrich et al., 2021). Additionally, the vegetation cover beneath the vines positively influences soil fertility, for example, it increases the SOC content (Fleishman et al., 2021; Marks et al., 2022), which can improve aggregate structure, though the extent of this effect varies by further soil properties that control the mechanisms of aggregate formation (Bonifacio et al., 2024). These factors in turn reduce soil erodibility, which also supports organic management practices in viticulture.

The argument against organic management practices is that the vegetation competes with the vines for water and nutrients (Celette et al., 2009; Dittrich et al., 2021), which can also negatively impact grape yields. Celette et al. (2005) found that vine vigour was reduced in a vineyard intercropped with tall fescue grass, resulting in a 30% lower grape yield compared to a conventional vineyard using chemical weed control. Although the cover crops in this study led to a reduction in available water for the vines, Celette et al. (2005) attributed this not only to water competition but also to competition for other soil resources such as nutrients or allelopathy effects. Similarly, Ruiz-Colmenero et al. (2011) demonstrated that permanent vegetation covers



that are not mowed can even reduce grape yield by up to 54% (with an average reduction of 40%). The extent of competition
likely depends on the climatic conditions of the vineyard location and is probably more pronounced in arid regions than in
humid ones, thus Trigo-Córdoba et al. (2015) showed that grape yield and quality in intercropped vineyards even improved
compared to a conventional vineyard in humid and sub-humid climates. Regardless of whether through soil erosion or planting
vegetation, there is a potential risk that the grape yield will be reduced. Therefore, in their review on cover crop management
and water conversation in vineyards, Novara et al. (2021) recommended the use of cover crops not only in humid but also in
drier areas due to their numerous benefits such as erosion control, increase of organic matter and improvement of soil fertility.
In dry areas, however, the choice of cover crop species and the timing of killing crops should depend on the average rainfall
(Novara et al., 2021).

An alternative to cover crops to combat soil erosion is a moss cover. Research in forests has shown that mosses reduce soil
erosion and surface runoff (Gall et al., 2022a; Gall et al., 2024a). Unlike cover crops, they do not require mowing, thereby
reducing maintenance efforts and costs. Furthermore, mosses thrive in conditions where vascular plants struggle, such as low
pH soils, steep slopes, or managed soils (Gall et al., 2022b; Corbin and Thiet, 2020). Some studies, have shown that they can
also enhance infiltration (Gall et al., 2024a), which depends on rainfall intensity and moss species (Tu et al., 2022), and prevent
soil evaporation (Thielen et al., 2021; Liu et al., 2022). However, there are also indications of opposite effects; for example,
in some cases, mosses have prevented infiltration (Li et al., 2022a), especially with low rainfall intensities (Tu et al., 2022),
and have increased soil evaporation (Li et al., 2022a), which could be detrimental to grape yields. Due to their potential
beneficial effects on the soil, planting mosses can be a promising new way for sustainable soil management (Gall et al., 2022b).
However, moss restoration over large areas is demanding and a growing research field. In recent years there have been
successful efforts to establish mosses in the field under different environmental conditions (Antoninka et al., 2020). For
instance, Bu et al. (2018) conducted a plot experiment (1 × 1 m) in a warm temperate environment in China and achieved a
moss cover of 85% using two dispersal methods (broadcast and spray), whereby this maximum cover was already obtained
after 30 days with spraying and after 60 days with broadcasting. For this, it was beneficial for moss growth to apply a nutrient
solution, maintain the soil water content at 15 to 25%, and provide moderate shade in summer. In comparison, Doherty et al.
(2020b) developed a moss-colonized burlap fabric, which was placed in the field for restoration, and was able to establish
itself when applied face-down despite drought during the observation period.

In addition, there have also been some encouraging experiments on the application of moss restoration strategies in practice,
for example in agriculture (Doherty et al., 2020a) or for post-fire recovery of forests (Grover et al., 2019; Grover et al., 2022),
although the moss cover remained small after restoration in all cases. This shows that there are still major challenges in the
development of sustainable technologies for moss restoration, which should be the focus of restoration research so that
application in practice over large areas becomes possible in the future. Furthermore, some areas of application, such as
viticulture, have not yet been considered for moss restoration, although the approach could be particularly promising for
erosion control in vineyards. This gap in research highlights the necessity for studies focused on the efficacy of mosses in
reducing soil erosion, particularly in vineyard environments.





This study aims to address this research gap by investigating the restoration of mosses in a temperate vineyard and evaluating their impact on surface runoff, sediment discharge, and soil water content of the topsoil. The following three hypotheses are formulated:

**(1) Mosses will begin to establish in the vineyard after being introduced to the field.**

**(2) Moss restoration reduces surface runoff and sediment discharge compared to cover crops and bare soil.**

**(3) Moss restoration increases soil water content in the upper 5 mm of the topsoil during rainfall simulations.**

With this research we want to contribute to the understanding of mosses as a practicable erosion control measure and provide practical knowledge for the management of vineyards to maintain soil fertility and prevent erosion.

## 2 Methodology

### 2.1 Study site

The study took place in a vineyard south of Fellbach, which is situated in southwestern Germany, approximately 10 km northeast of Stuttgart. The vineyard cultivates the vine variety Lemberger and the soil between the vines is continuously overgrown with cover crops. It is located at an altitude of 324 m above sea level at the foot of the Kappelberg (469 m above sea level) with flat slopes of 5° and is part of the Keuperbergland, which consists of Triassic hills stratified by sandstones, marlstones, and claystones (Geyer et al., 2023). According to the geological map (LGRB, 2022), the study site belongs to the Grabfeld-Formation, an alternating sequence of sulphate rocks (gypsum and anhydrite) and claystones in the upper Triassic series. A Rigosol as soil type was identified using a drill stick (Pürckhauer), which is typically formed in vineyards by deep, regular digging. Mixed samples of the topsoil (0-5 cm) were taken to describe main soil characteristics: Soil texture was medium clayey loam (Sand: 23.17%, Silt: 38.93%, Clay: 37.8%), the pH (CaCl$_2$) was 7.2, soil bulk density was 0.96 g m$^{-3}$, and SOC was 2.32%. A climate station in the immediate vicinity of the study site reveals an average annual temperature 11.5 °C between 2007 and 2023 while the average annual precipitation over the same period was 668.3 mm (Agrarmeteorologie Baden-Württemberg, 2024a).

### 2.2 Field methods

#### Treatment preparation

The treatments were established on February 17, 2022, directly within the vine rows at the study site. In total, there are three treatments: moss restoration (moss), bare soil (bare), and cover crops (grass), each with four replicates.

The bare treatment was set up by completely weeding the soil. Due to vegetation growth, this procedure had to be repeated before each rainfall simulation, although the soil surface was kept intact to avoid influencing soil erosion processes. This regular weeding maintained a minimal vegetation cover (2-20%), leaving only cut grass tufts and mosses.



The grass treatment utilizes the existing planted cover crops without additional preparation, which include mainly grasses but also other vascular plants and a few moss species underneath. Common species were for example *Lolium perenne*, *Trifolium repens*, *Trisetum flavescens*, and *Achillea millefolium* (identified using Jäger and Werner (2005)).

The moss treatment uses artificially grown moss mats with a mixture of mosses (*Amblystegium serpens* (Hedw.) Schimp., *Brachythecium rutabulum* (Hedw.) Schimp., *Funaria hygrometrica* Hedw., *Homalothecium lutescens* (Hedw.) Robins, *Oxyrrhynchium hians* (Hedw.) Loeske), produced by Hummel InVitro GmbH Stuttgart, Germany. Cultures of these moss species were propagated in hydraulic fluid in an in vitro environment and grown on jute fleece so that the moss mats can be easily rolled, transported and spread in a similar way to rolled turf. The moss treatment was installed by weeding the area,

cutting the moss mats to 40 ×40 cm, laying them on the bare soil, and securing them with a nail in each corner. Each moss mat was initially watered with 0.5 litres and periodically during dry, hot weather to ensure establishment.

For continuous temperature and soil moisture measurement, TMS-4 dataloggers (TOMST, Czech Republic) were installed on April 14, 2022 in all treatments. Each TMS-4 has three temperature sensors located 6 cm below the soil surface, 2 cm, and 15 cm above the soil surface, while the moisture sensor measures to a soil depth of approximately 14 cm (Wild et al., 2019).

The data was collected for 21 months for analysis.

**Rainfall simulation**

To analyse the effect of moss restoration on soil erosion and soil water content, three rainfall simulations were conducted at three measurement times: April 13-14, June 14-15, and October 24-25, 2022. These dates were chosen to study soil erosion across seasons and monitor the development of the moss mats. The first and second simulations also assessed the impact of

vine foliage on soil erosion: vines were leafless in April but nearly fully leafed by June. Surface runoff and sediment discharge were measured using micro-scale runoff plots (ROPs, 40 x 40 cm; cf. (Seitz, 2015)) for each treatment. The Tübingen rainfall simulator, modified with a pavilion for wind protection and an adjusted fall height of 2 meters, was used. It featured a Lechler 490.808.30.CE nozzle set to a rainfall intensity of 45 mm h$^{-1}$ for 30 minutes. Runoff and sediment were collected in 1 liter sample bottles. Soil water content was measured with biocrust wetness probes (BWP) from UP GmbH, Cottbus, Germany for

each rainfall simulation. Therefore, BWPs were placed in the upper 5 mm of the soil surface underneath the respective vegetation. To determine vegetation cover with a photogrammetric survey, perpendicular photos of all ROPs were taken with a digital compact camera (Panasonic DC-TZ91, Osaka, Japan) during each rainfall simulation. Afterwards, the photos were analysed with the grid square method using a digital grid overly with 100 subdivisions (Belnap et al., 2001). For each subdivision bare soil and vegetation covers were separated by hue distinction.

**2.3 Climatic conditions after treatment preparation and before rainfall simulations**

To evaluate the progress of moss restoration, the climatic conditions from the preparation of the treatments to the first rainfall simulation must be taken into account, as shown in Figure 1, which was created based on the data of the climate station of Fellbach (Agrarmeteorologie Baden-Württemberg, 2024a, b). A total of 51.1 mm of precipitation was recorded and the average




air temperature was 6.65 ± 0.15 °C for the period 48 days from the start of the moss restoration until one week before the first

rainfall simulation (February 17th 2022 – April 5th 2022). In February and March 2022, precipitation sums were especially low

compared to the respective monthly long-term averages of the region (1961 to 1990 climate station in Waiblingen; February

2022: 34.3 mm, long-term average for February: 48.8 mm; March 2022: 20.3 mm, long-term average for March: 48.8 mm),

while average air temperature was especially high (February 2022: 6.4 °C, long-term average for February: 1.5°C; March

2022: 7.2 °C, long-term average for March: 5.1°C). Figure 1 also shows that high daily sums of global radiation were achieved

at some days, which is also reflected in increased hours of sunshine compared to the long-term average (February 2022: 85

hours, long-term average for February: 80 hours; March 2022: 199 hours, long-term average for March: 124 hours). For this

reason, the average values for relative humidity were below 50% on some days.

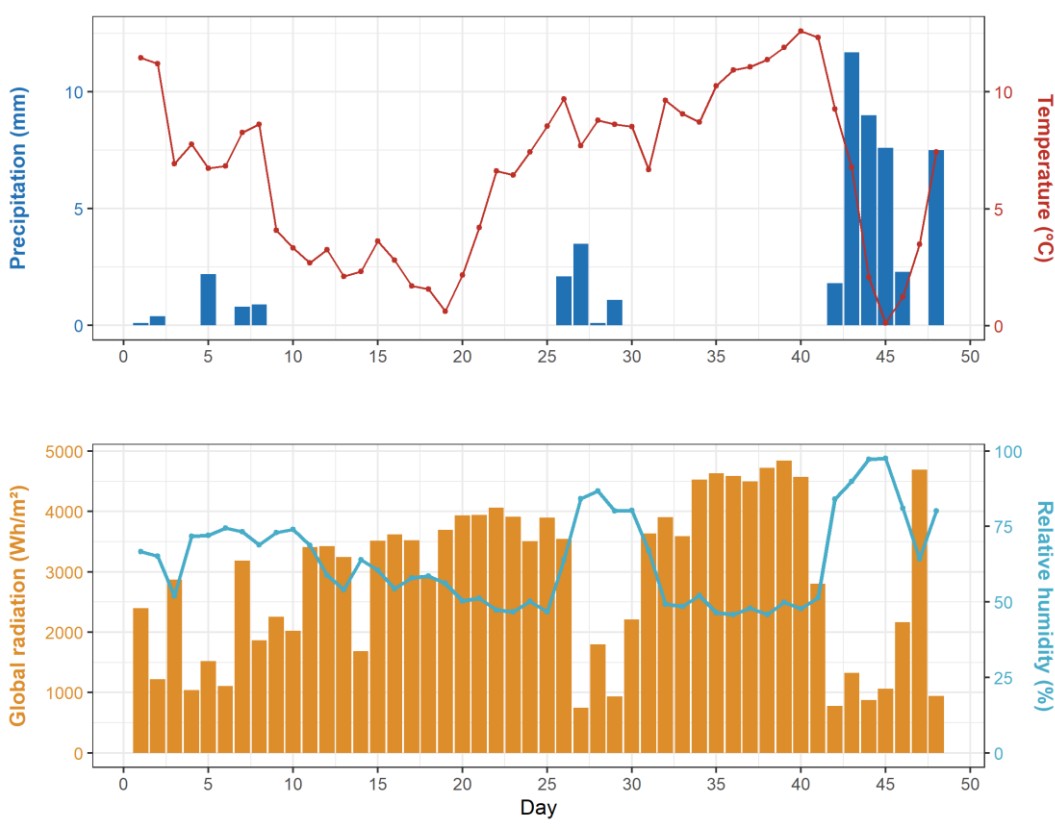

**Figure 1: Climate diagram for Fellbach with daily sum of precipitation (mm), average air temperature (°C), daily sum of global**
**radiation (Wh m⁻²) and average relative humidity (%). Displayed are the 48 days from the start of the moss restoration until one**
**week before the 1. rainfall simulation from February 17th 2022 to April 5th 2022 (Agrarmeteorologie Baden-Württemberg, 2024a, b)**

In addition, the climatic conditions prior to the rainfall simulations can be useful for assessing the time course of soil moisture.

Figure 2 therefore shows the week before each rainfall simulation with the two days on which the simulations were carried

out, i.e. a total of 9 days. In April, the sum of precipitation amounted to 33.1 mm during this time period, with 25 mm occurring



on one day alone. The average temperature was 10.80 ± 0.38 °C, dropping to a minimum of 4.64 °C on day 4 and rising steadily thereafter. In contrast, the total sum of precipitation in the week before the second rainfall simulation in June was much lower at 10.4 mm and distributed over five days. The average temperature was considerably higher at 19.44 ± 0.33 °C, whereby the changes in the daily average temperature were smaller compared to April. In June, the total sum of precipitation in the week before the third rainfall simulation was 14 mm, which was distributed over four days. Between the first and second day of the

rainfall simulation, a rainfall event of 6.9 mm occurred in the evening. The average temperature was 15.84 ± 0.21 °C, with only slight fluctuations in the daily average.

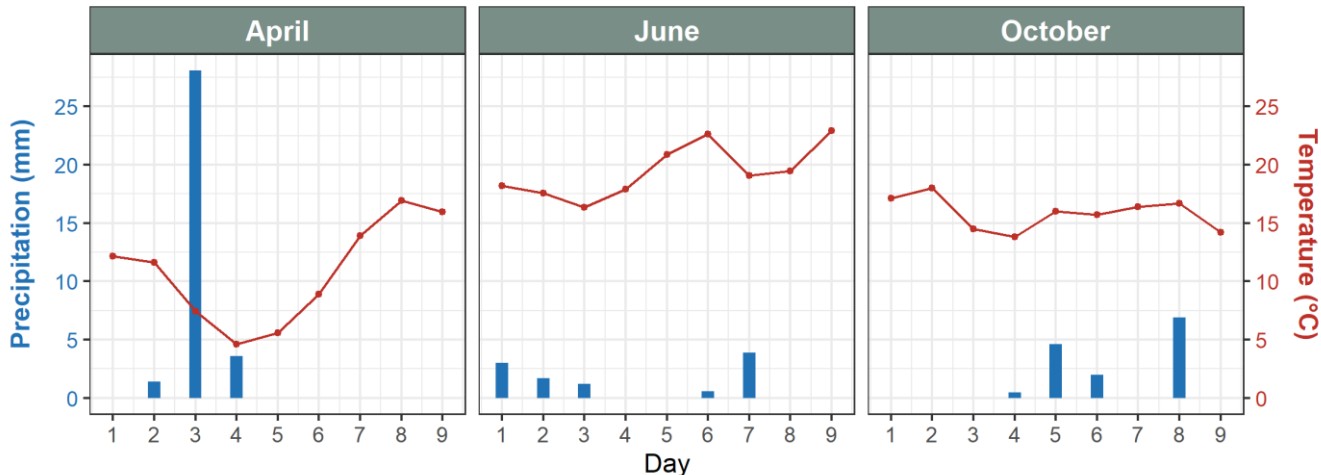

**Figure 2: Climate diagrams for Fellbach with daily sum of precipitation (mm) and average air temperature (°C). The seven days before and the two days of rainfall simulation (day 8 and 9) are shown (April 6th – April 14th 2022; June 7th – June 15th 2022, October**
**17th – October 25th 2022, Agrarmeteorologie Baden-Württemberg (2024a, 2024b))**

**2.4 Laboratory analysis and BWP calibration**

After the rainfall simulations, the amount of surface runoff was determined using the sample bottle scales. Surface runoff samples were then evaporated at 40 °C in a compartment drier to weigh the eroded sediment. The following basic soil properties were determined using the mixed soil sample collected prior to the first rainfall simulation: grain size distribution with an x-

ray particle size analyser (Sedigraph III, Micromeritics, Norcross, GA, USA); soil pH in a 0.01 M $CaCl_2$ solution with a pH meter with Sentix 81 electrodes (WTW, Weilheim, Germany); soil organic carbon (SOC) with an elemental analyser (Vario EL II, Elementar Analysensysteme GmbH, Hanau, Germany); soil bulk density in 100 cm³ core samples using the mass-per-volume method (Blake and Hartge, 1986).

Since the electrical conductivity recorded by the BWP is temperature-dependent, a correction was applied to adjust all

measurements to 25 °C, following Weber et al. (2016). Furthermore, a simplified calibration procedure, as suggested by Weber et al. (2016), was used to calibrate the BWP values from electrical conductivity (mV) to a gravimetric water content (g g$^{-1}$).



Therefore, soil samples were weighed in 100 cm³ core cutters in both water-saturated and dry (40 °C) conditions to establish linear calibration functions for the minimum and maximum water content of each soil substrate.

## 2.5 Data analysis

Data analysis was conducted using R software version 4.0.4 (R Core Team, 2021). Normality was tested with the Shapiro-Wilk test prior to all statistical tests, while homoscedasticity was verified with Levene's test. As our data were not normally distributed and not homoescedastic, the Kruskal-Wallis test was used to screen for significant differences. Dunn's test was applied as a posthoc test, as it allows for a check of significant differences with a small sample size. The 10-second time series of the water content were analysed using minute mean values, with the Wilcoxon rank sum test applied for differences between

treatments and the Dunn's test for differences between measurement times within treatments. Significant differences were postulated in all cases at $p < 0.05$. For all mean values described, the standard error was also given (mean ± standard error). Spearman pairwise correlation analysis was performed to describe relationships between different parameters. The colours selected for all figures are from the R package "wesanderson" (Karthik et al., 2018).

## 3 Results

### 3.1 Development of moss restoration

The percentage vegetation cover per ROP (Figure 3) was determined for each measurement time and is summarized in Table 1. The bare treatment has the lowest vegetation cover for all measurement times, whereby in April and June the remaining vegetation is characterized by cut grass tufts, and in October some mosses could not be removed without damaging the soil surface. In the grass treatment, the vegetation cover was 100% for all measurement times, although a noticeably lower growth

height of the grasses can be seen in April compared to June and October. The moss treatments dried out considerably in April and June and in both measurement times the jute fleece under the mosses is still clearly visible. Additionally, the moss cover has noticeably decreased from April to June. Only in October the jute fleece under the mosses is completely decomposed, the moss cover has increased again and the mosses appear green and vital.






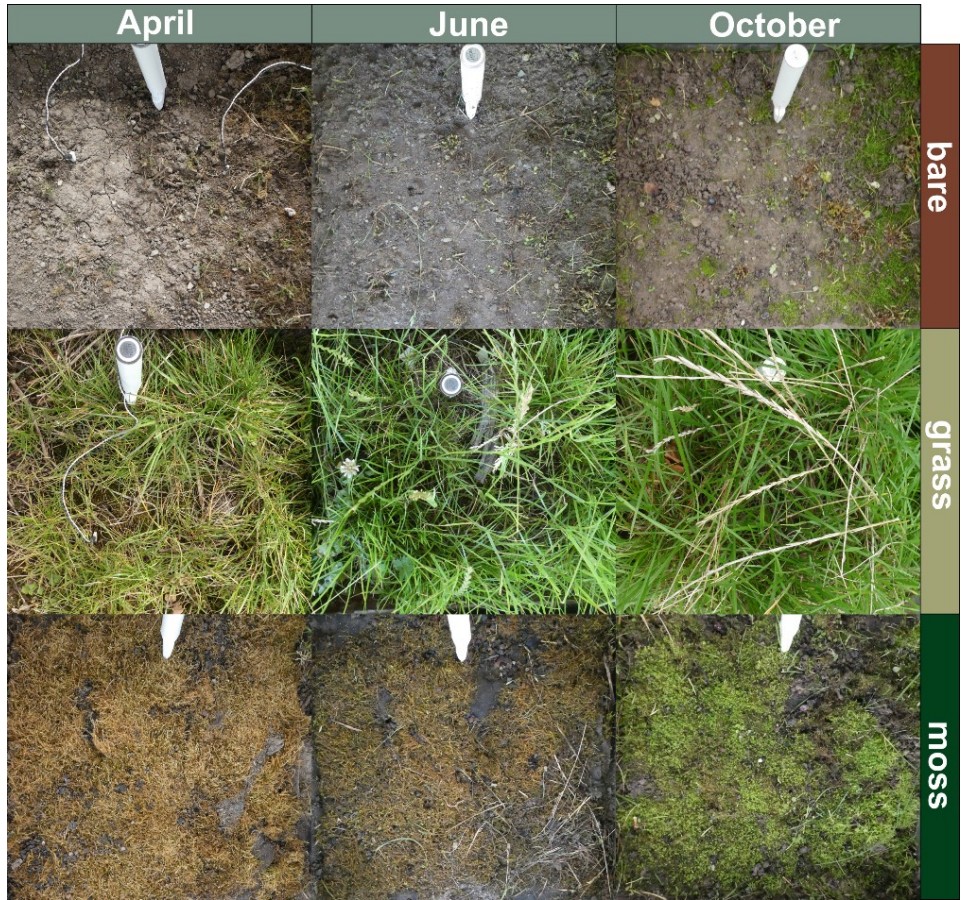

**Figure 3: Exemplary development of vegetation cover over time in one runoff plot for all three treatments, respectively.**

**Table 1: Vegetation cover in % for all runoff plots (ROPs) and treatments in April, June, and October**

| ROP number | bare | | | grass | | | moss | | |
|---|---|---|---|---|---|---|---|---|---|
| | April | June | Oct. | April | June | Oct. | April | June | Oct. |
| **1** | 4 | 6 | 10 | 100 | 100 | 100 | 91 | 81 | 82 |
| **2** | 3 | 2 | 8 | 100 | 100 | 100 | 83 | 67 | 82 |
| **3** | 10 | 6 | 20 | 100 | 100 | 100 | 91 | 87 | 94 |
| **4** | 6 | 4 | 7 | 100 | 100 | 100 | 96 | 60 | 65 |
| **Mean** | **5.75** | **4.50** | **11.25** | **100** | **100** | **100** | **90.25** | **73.75** | **80.75** |

**3.2 Effect of moss restoration on surface runoff and sediment discharge**

Taking the mean for all measurement times, it can be observed that both the moss and the grass treatment significantly reduce surface runoff (moss: $6.27 \pm 1.92$ L m$^{-2}$, $p < 0.01$; grass: $6.99 \pm 2.27$ L m$^{-2}$, $p < 0.01$) compared to the bare treatment ($21.92 \pm$





2.52 L m$^{-2}$), which corresponds to a decrease in surface runoff of 71.4% and 68.1%, respectively. Even though the moss treatment has a slightly lower mean surface runoff than the grass treatment, no significant difference is detected between the two treatments. A separate consideration of the measurement times shows that the surface runoff is influenced by seasonality

(Figure 4). Especially for the moss treatment, there is a significant increase in surface runoff between April (0.91 ± 0.20 L m$^{-2}$) and October (10.39 ± 4.12 L m$^{-2}$, $p < 0.05$). Additionally, surface runoff for the moss treatment is significantly lower than for the bare treatment in April, while the reduction in surface runoff is only significant for the grass treatment in June. In October, no significant difference in surface runoff is observed between the three treatments.

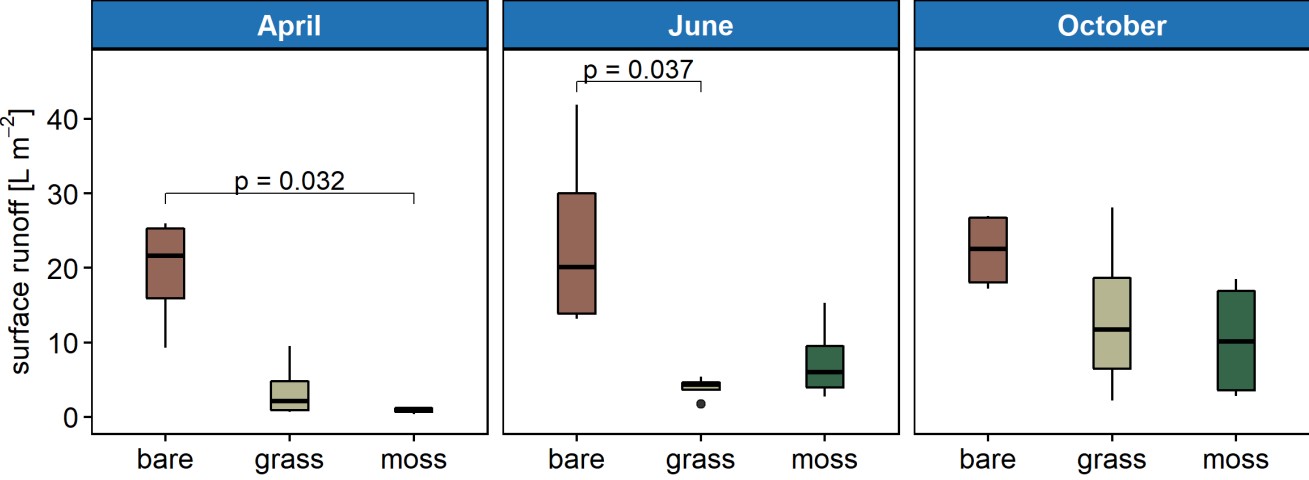

**Figure 4: Surface runoff [L m$^{-2}$] for three treatments and three measurement times (*n = 4*). Lines within boxplots represent median values, while bottom and top of the boxplot show the first and third quartiles. Whiskers extend up to 1.5 times the interquartile range (IQR) of the data. Outliers are defined as more than 1.5 times the IQR and are displayed as points. The p values presented indicate significant differences between treatments and are based on the Dunn`s test.**

On average for all measurement times, sediment discharge is highest for bare treatments (139.49 ± 34.57 g m$^{-2}$) with a

significant reduction in the grass treatment (17.21 ± 3.91 g m$^{-2}$, $p < 0.001$) and the moss treatment (33.74 ± 13.08 g m$^{-2}$, $p < 0.01$), corresponding to a reduction in sediment discharge of 87.7% and 75.8%, respectively. However, there is no significant difference in sediment discharge between grass and moss treatment. As with surface runoff, the influence of seasonality is also visible in sediment discharge separated by measurement time (Figure 5), whereby the significant increase in sediment discharge in the moss treatment between April (1.31 ± 0.73 g m$^{-2}$) and June (83.25 ± 24.12 g m$^{-2}$, $p < 0.01$) is

particularly noteworthy. In April, the moss treatment leads to a significant reduction in sediment discharge compared to the bare treatment, while in June and October, the grass treatment produces significantly lower sediment discharge than the bare treatment and not the moss treatment.





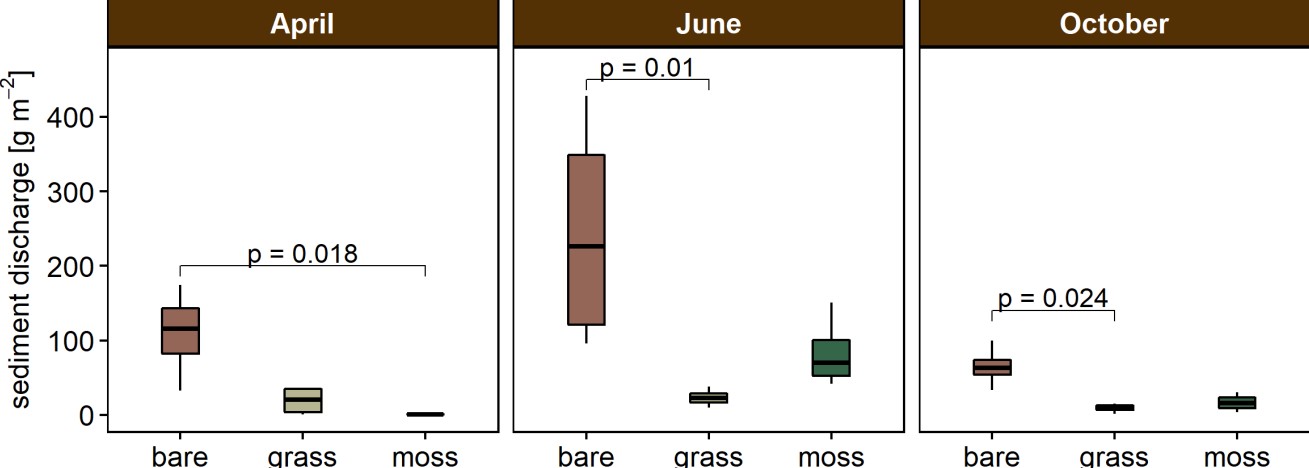

**Figure 5: Sediment discharge [g m⁻²] for three treatments and three measurement times (*n = 4*). Lines within boxplots represent**
**median values, while bottom and top of the boxplot show the first and third quartiles. Whiskers extend up to 1.5 times the interquartile range (IQR) of the data. Outliers are defined as more than 1.5 times the IQR and are displayed as points. The p values presented indicate significant differences between treatments and are based on the Dunn`s test.**

### 3.3 Effect of moss restoration on soil water content

The results of the water content measurements in the first 5 mm of the topsoil are shown in Figure 6 for all rainfall simulations
and treatments. Depending on the season in which the rainfall simulation was carried out, differences in the course of the water content between the three treatments can be recognized.

In the first rainfall simulation in April, water content in bare soil ($0.181 \pm 0.014$ g g⁻¹) is significantly lower than in both grass ($0.235 \pm 0.009$ g g⁻¹, $p < 0.05$) and moss treatments ($0.240 \pm 0.018$ g g⁻¹, $p < 0.05$) within the first minute. There is no significant difference between grass and moss treatments. During the fifth and sixth minutes, water content remains lower in
bare compared to moss treatments (both $p < 0.05$). After this period, no significant differences are observed between the three treatments for the rest of the rainfall simulation.

In contrast, the second rainfall simulation in June shows higher water content in bare soil ($0.238 \pm 0.009$ g g⁻¹) than in moss ($0.172 \pm 0.019$ g g⁻¹, $p < 0.05$) and grass treatments ($0.155 \pm 0.010$ g g⁻¹, $p < 0.001$) within the first minute. Similarly, bare soil maintains higher water content compared to grass treatments in the second and third minutes. After these initial minutes,
no significant differences are detected among the treatments.

The third rainfall simulation in October presents a different scenario: During the first minute, water content in bare soil ($0.333 \pm 0.001$ g g⁻¹) is higher than in moss ($0.309 \pm 0.001$ g g⁻¹, $p < 0.001$) and grass treatments ($0.305 \pm 0.003$ g g⁻¹, $p < 0.001$), with no significant difference between moss and grass. However, from the second minute onwards, significant differences are consistently observed among all treatments for the duration of the simulation. Bare soil exhibits the highest
water content, followed by grass, and the lowest water content is found in moss treatments.



In addition, there are notable seasonal differences in water content across the treatments. Figure 6 illustrates that water content is highest in October for all treatments, followed by April, and is lowest in June, and these visible differences are statistically significant in most cases. The only exceptions are that no significant differences are found in the bare treatment between April and June in the period of 2 to 10 minutes, in the moss treatment between April and October in the period of 4 to 9 minutes
(and some individual minutes thereafter), and in the moss treatment between April and June in the period of 10 to 30 minutes.

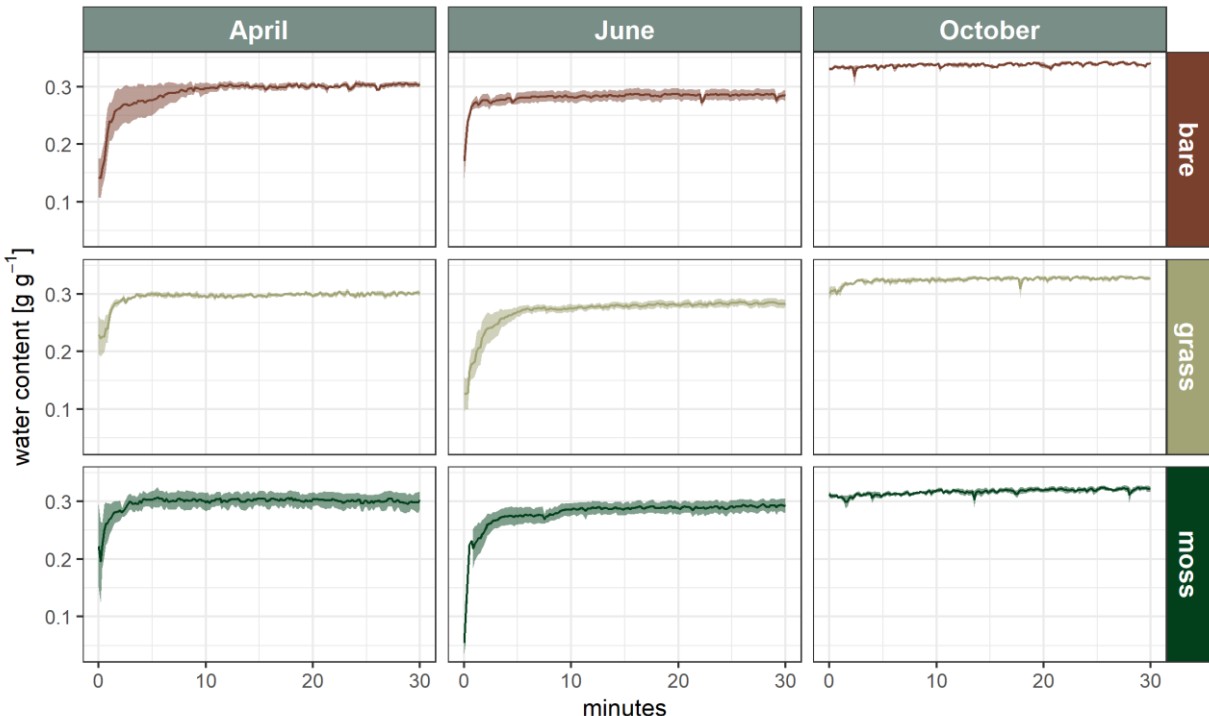

**Figure 6: Temporal dynamics of soil water content [g g⁻¹] of each treatment and measurement time. The mean values of the 10-second measurements are shown as a line and the standard errors as a ribbon (_n = 4_). Soil water content was measured with biocrust wetness probes (BWP) in the first 5 mm of the topsoil.**

**4 Discussion**

**4.1 Development of moss restoration**

The moss mats have established themselves in the vineyard more slowly than we originally expected. This can be attributed primarily to the atypical climatic conditions observed during the restoration period (Figure 1). The composition of the moss mats includes species that thrive in a variety of habitats from shaded forest floor to open grassland (Nebel et al., 2000; Atherton
et al., 2010). Moss species growing in these environments can generally tolerate at least occasionally dry periods, but they are not known to be particularly desiccation-tolerant (Proctor et al., 2007). Especially, for the initial growing and acclimatisation of the mosses in the vineyard, a high water requirement was expected and based on historical climate data, we assumed that



March would provide sufficient rainfall for moss establishment (Agrarmeteorologie Baden-Württemberg, 2024a). Instead, the mosses experienced substantial stress due to the unusually dry and warm weather, which led to desiccation and a subsequent
decline in moss cover during the summer months. Similar findings from other studies emphasize that water availability is a critical factor for the success of moss restoration efforts (Grover et al., 2022; Doherty et al., 2020b). Nevertheless, the great resistance of the moss species involved led to a final establishment success. Even though soil protection was less effective in the summer months, a vital and healthy moss cover was re-established from October onwards and fulfilled the expected ecosystem functions. This can be seen as an advantage of moss mats in changing extreme climatic situations.
There have already been promising approaches to moss restoration that employ adaptive management strategies to account for climatic variability. Bu et al. (2018) have shown, for example, that the rapid restoration of moss worked well with sufficient irrigation (70 litre per plot of 1 x 1 m in 75 days, in addition to natural rainfall) and shading. However, besides sufficient water supply and temperature, many more factors such as soil pH, nutrients, calcium carbonate content, or soil texture play an essential role for moss growth (Glime, 2017). This suggests that it may be necessary to develop species-specific solutions for
moss restoration, taking into account the major constraints of the species involved (Adessi et al., 2021).

## 4.2 Effect of moss restoration on surface runoff and sediment discharge

Overall, surface runoff was strongly reduced by moss restoration and cover crops compared to bare soil, although the reduction in runoff was slightly higher for moss restoration, albeit not significantly. The runoff-reducing effect of mosses has already been confirmed in several studies (Xiao et al., 2015; Tu et al., 2022), with the extent of the reduction varying widely from
28.8% in Juan et al. (2023) to 91% reduction compared to bare soil in Gall et al. (2024a). However, Gall et al. (2024a) could show that runoff reduction was also strongly influenced by desiccation cracks. In contrast to our results, Bu et al. (2015) measured a runoff reduction of 37.3% by moss-dominated biocrusts compared to bare soils, while two different grass species alone (*Stipa bungeana* Trin. and *Caragana korshinskii* Kom.) reduced surface runoff even more (58.5% and 90.1%, respectively). A combination of mosses and the two grasses increased the runoff reduction by just 7.4% and 5.7%, respectively.
This wide range of runoff reduction also shows that, in addition to moss cover, many other factors influence surface runoff such as antecedent soil moisture, aggregate structure, soil texture, and many more (Le Bissonnais and Singer, 1993; Le Bissonnais et al., 1995; Knapen et al., 2007).

In addition, the reduction in runoff varied seasonally, particularly with regard to the restoration of moss, which decreased steadily from April to October. This can be ascribed to the decline in moss cover on the one hand, but also to the delayed
decomposition of the jute mats on the other. We had originally assumed that the surface runoff would decrease as soon as the mosses had established themselves at the site, but on average the highest surface runoff was measured in October. One reason for this could be that in October, despite the complete establishment of the mosses, the soil coverage was lower than in April. In addition, it could be that the jute material itself has contributed substantially to runoff reduction, as jute nets are also often used as a geotextile for soil protection and their runoff and erosion-reducing effect has been demonstrated in several studies



(Bhattacharyya et al., 2010; Mitchell et al., 2003). Here the measurement was influenced by the low decomposition rates caused by the summer drought.

Moss restoration strongly reduced sediment discharge in the vineyard, but erosion was more strongly reduced by cover crops, although the difference was not significant. Similarly, the study by Bu et al. (2015) showed that two different grass species reduced soil erosion more compared to bare soils (*Stipa bungeana* Trin. by 95.9% and *Caragana korshinskii* Kom. by 99.5%)

than moss-dominated biocrusts (erosion reduction by 81.0%). In contrast, Gall et al. (2022a) found that moss-dominated runoff plots reduced sediment discharge by 77%, while runoff plots dominated by vascular vegetation just mitigated sediment discharge by 59%, albeit the difference was not significant. However, it is important to distinguish between moss-dominated biocrusts and moss-covered soils (Weber et al., 2022), as these two types of mosses can likely have different effects on runoff and erosion control due to their very different structure. While biocrusts form in the upper millimetres of the soil and create

an encrusted surface, with only a small part of their biomass protruding above the soil surface, mature moss covers grow mainly on top of the soil surface, and depending on the species, they are not even attached to the soil and create thick mats or lawns (Weber et al., 2022). Due to the diverse life forms of mosses (Bates, 1998), it is also possible that the impact on runoff formation and sediment discharge varies from species to species (Tu et al., 2022; Gall et al., 2024a; Thielen et al., 2021).

The seasonal fluctuations in sediment discharge, especially between April and June in the moss restoration, can be attributed

to the fact that the moss cover decreased significantly during this period and that the vines were foliated in June, which was not the case in April. So far, only a few studies have examined the impact of leaves and species-specific plant traits on soil erosion. For example, Seitz et al. (2016) found that in a young subtropical forest in China, trees influence soil erosion based on species and their respective functional traits, whereby particularly high crown cover and leaf area index significantly controlled soil erosion. Investigating species' functional traits is crucial, as they greatly affect throughfall kinetic energy,

consequently affecting splash erosion (Seitz et al., 2016; Goebes et al., 2016; Goebes et al., 2015).

However, the effect of individual trees or tall plants, such as vines, on soil erosion is still unclear, as to our knowledge no study deals with the effect of vine leaves on soil erosion. The leaf blades of the investigated vines are pointed at the front, which may lead to the formation of particularly large droplets that result in a higher splash effect. For instance, Nanko et al. (2013) showed that leaf geometry is, among other things, decisive for leaf drip drop size distribution. Additionally, a further splash

effect became visible on bare soils, as we found drop impact holes on the soil surface after the rainfall simulations. We suspect that large drops have repeatedly formed at structurally-mediated woody surface drip points, leading to this severe form of erosion, which was recently reported by Katayama et al. (2023), who described these concentrated points as hotspots of soil erosion in forests.

**4.3 Effect of moss restoration on soil water content**

During the rainfall simulations in April and June, there were no differences in soil water content in the upper 5 millimetres between bare soil, cover crops, and moss restoration, but the initial soil water content in bare soil was lower in April and higher



in June than in both vegetation-covered soils. This can probably be explained by the climate conditions before the rainfall simulations (Figure 2).

In April, there was no natural precipitation and air temperature steadily increased three days before the rainfall simulation, so that the bare soil desiccated on the upper soil surface, while cover crops and moss restoration prevented desiccation of the soil surface. Similar results were obtained by Thielen et al. (2021), who found that mosses prevent the desiccation of the soil and mitigate soil evaporation. However, other studies suggest that moss-dominated biocrusts increase evaporation (Li et al., 2022a; Li et al., 2022b), although this may be due to the encrustation, which does not occur in our moss restoration. Interestingly, Chen et al. (2019) discovered that mosses actively regulate evaporation to keep their temperature below a threshold of 30°, although they are theoretically unable to do so due to their poikilohydric nature (Glime, 2017). Despite some initial studies, the influence of mosses on soil evaporation remains enigmatic, highlighting the need for further research on this topic.

In June, there was a small natural rainfall event of less than 5 mm one day before the rainfall simulation, which may have resulted in the bare soil being wetter than the two vegetation-covered soils due to rainfall interception. Interception plays a decisive role, especially in the case of small rainfall events, as water may not reach the soil (Dunkerley, 2000). As mosses can absorb a lot of water (Wang and Bader, 2018; Thielen et al., 2021), it can be assumed that the interception effect of mosses is very high. For example, Price et al. (1997) found that moss covers were able to retain 16.8 mm of precipitation, corresponding to approximately 21% of the precipitation input in a boreal forest. However, the initial difference in the soil water content already disappeared after one minute of rainfall simulation in our experiment, so no difference between bare soil, cover crops, and moss restoration could be found.

In contrast, there was a clear difference in soil water content between the treatments in October, with bare soil exhibiting the highest and moss restoration the lowest water contents. This indicates that the type of soil cover has a greater influence on the soil water content in October than in other seasons. Such seasonality of soil water content was also observed in a study by Marques et al. (2020), comparing cover crops with conventional tillage management in a Spanish vineyard. However, the cover crops here led to an increase in soil water content at a depth of 10 cm in autumn on wetter soils, while in spring and summer soil water content under cover crops was considerably lower compared to conventional tillage management. This is an opposite trend compared to our results, which can probably be attributed to different climatic conditions and soil characteristics of the research sites.

In addition, the water content in all treatments is highest in October, followed by April and lowest in June. Such seasonal differences in soil water content were also measured by Siwach et al. (2021) at three different sites in the temperate forest zone of the Garhwal Himalayas, whereby the water content in winter was higher under moss than in the soil without moss and in the monsoon season exactly the other way round. These different responses in various seasons highlight the need to consider seasonal variations in soil and vegetation management practices.



**5 Conclusion**

This study investigated moss restoration in a temperate vineyard, evaluating its impact on surface runoff, sediment discharge,
and soil water content at the soil surface. The conclusions regarding our hypotheses are as follows:

1.  Due to unexpected dry weather and insufficient watering, the mosses initially dried out after restoration in February and
    recovered in October, albeit with less cover. Therefore, future moss restoration projects should incorporate flexible
    planning to address climatic fluctuations such as selecting more desiccation-tolerant species or providing additional
    irrigation during critical periods. Developing species-specific solutions considering major constraints may be also
necessary.

2.  The strongest reduction in surface runoff was achieved by moss restoration (71.4%), and was slightly higher than the
    reduction by cover crops (68.1%). Runoff reduction varied seasonally, decreasing steadily from April to October due to
    declining moss cover and delayed jute mat decomposition. Moss restoration significantly reduced sediment discharge by
    75.8% compared to bare soil, but cover crops reduced erosion more (by 87.7%). The seasonal fluctuations in sediment
discharge, especially from April to June, are due to the decline in moss cover and the foliage of the vines in June, as
    concentrated canopy drip points have formed on the leaves and woody surfaces of the vines, which considerably increase
    erosion.

3.  During the rainfall simulations in April and June, soil water content in the top 5 millimetres was similar across bare soil,
    cover crops, and moss restoration. However, bare soil had lower initial soil water content in April and higher in June and
October compared to vegetation-covered soils. In October, bare soil had the highest and moss restoration the lowest water
    content.

This study demonstrated that moss restoration can reduce soil erosion and surface runoff without decreasing near-surface soil
water content during the dry summer months in temperate vineyards. With improved application methods, mosses could
effectively limit soil erosion under vine rows, particularly in steep vineyards or those with challenging parent material that are
difficult for vascular plants to colonize. Additionally, mosses require minimal maintenance once established, as they do not
need mowing. This characteristic makes them particularly suitable as ground cover under vines, where mowing is impractical
and herbicides are commonly used. Consequently, successful moss restoration in viticulture has the potential to reduce the
environmentally harmful application of herbicides, though further research is necessary to realize this potential.

**Acknowledgements**

This research would not have been possible without the provision of the experimental fields by Marc Jäger, the owner of the
vineyard, and we would like to thank him for his great contribution and support. We also thank Gert Joachim Aldinger and the
Fellbacher Weingärtner eG for the opportunity to use their facilities during rainfall simulations. Furthermore, we sincerely
thank Carla L. Webber, Julia Dartsch, Nicolás Riveras-Muñoz, Larissa Werner, Caspar Hollmann, and all students of the





course GEO 51 in winter semester 2022/23 for their help with field and lab work. We also appreciate Sabine Flaiz, Rita Mögenburg and Peter Kühn for their lab work support. During the preparation of this work, ChatGPT-4 by OpenAI was used for refining language and grammar of the manuscript in individual cases.

**Financial support**

This research has been funded by the Deutsche Forschungsgemeinschaft (DFG SE 2767/2-1, "MesiCrust") and the Federal Ministry of Food and Agriculture / Federal Ministry for the Environment, Nature Conservation, Nuclear Safety and Consumer Protection via Fachagentur Nachwachsende Rohstoffe e.V. (FNR 2220WK67A4, "AnKliMoos"). We acknowledge support from the Open Access Publishing Fund of the University of Tübingen.

**Data availability**

The dataset compiled and analysed in this study is available on figshare (Gall et al., 2024b).

**Code availability**

The codes used in this study are available upon request.


**Author contribution**

CG, StS and MN designed the experiment. SO, CG, and StS carried out field measurements. SO was responsible for laboratory analyses, while SO and CG conducted data analyses. CG prepared the manuscript with contributions from all other co-authors.

**Competing interests**

The contact author has declared that none of the authors has any competing interests.

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
