# Peer review of "Effects of moss restoration on surface runoff and initial soil erosion in a temperate vineyard"

_EGUsphere, 2024_

## Author Comment (AC1)

**Response to Anonymus Referee #1 (RC1) to**

**preprint egusphere-2024-2504: "Effects of moss restoration on soil erosion and soil water content in a temperate vineyard"**

Thank you very much for your thoughtful and positive review. We really appreciate the time and effort you put into your valuable and helpful feedback. Please find the answers to your comments in the table below. We will also incorporate them into our manuscript.

| Reviewer comments | Authors responses |
|---|---|
| *"This manuscript proposes moss restoration as a strategy for reducing erosion in vineyards, and presents the results of a plot-scale field experiment implementing the technique. Moss restoration, bare soil, and grass cover crop are compared, and the rates of runoff, erosion, and infiltration are measured using three simulated rainfall events. The paper demonstrates potential for moss to reduce runoff and erosion, and also identifies gaps for future research. Overall, the quality of the writing is very strong, and the content is presented very clearly. The introduction is particularly concise and comprehensive, to the great benefit of the manuscript."* | Thank you for the overall positive feedback! |
| *"While the central findings of runoff and erosion are solid, the manuscript could still be improved with changes to the data presentation and discussion. The most serious shortcoming of this manuscript is the conditions of the field trial itself. The study site previously had mixed grass and moss in the interr-row, not fallow or cultivated soil as in standard practice. Thus the initial soil conditions may have been different than typical vineyard soils. As the manuscript describes, the weather during the beginning of the trial was unusually warm and dry, leading to challenges establishing the moss matts. Combined with the fact that the "moss" treatment was implemented by spreading moss-impregnated burlap over the soil, it is difficult to disentangle the effects of the burlap from the moss, especially in the first two rain simulation events."* | We are aware of the difficulties to disentangle the effects of the moss itself and the underlying jute fleece and we mentioned this already in lines 318 – 321 of the original manuscript. However, we will discuss this issue in more detail at this point.

Regarding your comment on the initial soil conditions in the vineyard, we would like to point out that our rainfall simulation experiments did not take place in the inter-row (between the rows of vines) but in-row (between the vines within a row). In this area of the vineyard, the soil is typically not tilled, but is treated with herbicides to control plant growth so that the soil surface remains undisturbed. We will include photos of the vineyard in the revised manuscript to help readers visualize the location of our treatments in the in-rows. |
| *"The manuscript could be improved by discussion of how widely-applicable moss could be to vineyards, given the range of aridity experienced in wine-growing regions."* | We agree that discussing the applicability of moss restoration to vineyards across diverse wine-growing regions is important, particularly given the variation in aridity. In the revised manuscript, we will expand the discussion to address this point. |
| *"The experimental design and results clearly demonstrate the runoff and erosion rates for the various treatments, but the treatment of* | Additionally, we have a continuous dataset with air temperature (15 cm above soil surface), soil temperature (2 cm and 6 cm |

| | |
|---|---|
| *long-term soil moisture is lacking in the present manuscript. The introduction identifies that concern about water competition is a primary barrier to implementing cover crops, but the data presented in the manuscript does not effectively characterize the water use of the three treatments. Soil moisture data is only presented for 3 points in time, and at 5 millimeter depth. This zone is particularly sensitive to recent weather, and is not necessarily representative of the water availability to roots. The formulation of hypothesis 3 and presentation of 30-minute time series of water content during simulated rain does not capture the important hydrological properties of the treatment.”* | below soil surface), and soil water content (approx. 14 cm below soil surface) captured every 5 minutes for 21 months (described in lines 137-140 of the original manuscript). These data are currently being analyzed and are too extensive to be included in this manuscript, which is why we decided to publish these data later in a second manuscript. Based on your comment, we decided to focus this manuscript on the effects of moss restoration on surface runoff and soil erosion, and also on the measure of using moss mats in vineyards for erosion control itself. We will therefore move the information on the soil water content in a reduced form to the supplementary material. |
| *“Hypotheses 1 and 2 are very strong, but hypothesis 3 seems less strongly motivated. From the data collected for soil moisture at 5mm, perhaps there could be some analysis of infiltration rate at the surface that would be interesting. But I would suggest that hypothesis 3 and the associated data could be moved to the supplement, and the discussion could be relatively unchanged. This could also allow the information in figures 1 and 2 to be streamlined or moved to the supplement.”* | As already mentioned before, we will exclude hypothesis 3 from our manuscript and show the respective data in the supplementary material. However, we will keep at least Figure 1 in the manuscript, as the information about the weather conditions before the moss mats were installed is important to understand their development at the study site. |
| *“The introduction very nicely identifies the state of knowledge for moss restoration and erosion. However, I think the readability would be improved by specifically acknowledging knowledge gaps, especially those that will be addressed in this study.”* | Thank you for this suggestion. We will revise the introduction to explicitly highlight the key knowledge gaps, such as the very limited knowledge of moss applications in agricultural settings like vineyards and their potential to mitigate soil erosion under different soil management practices. |
| *“The introduction rightly identifies water consumption as a major risk of moss/cover crops, but the experiment mostly does not address this topic. I think this would be great to identify as a knowledge gap in the discussion for future work.”* | Following your comment, we will shorten the information on the risk of water consumption by mosses and cover crops in the introduction and move it to the discussion, where we will go into more detail and highlight this aspect as a knowledge gap that should be addressed in the future. |
| *“The applicability of moss to vineyards should be better addressed in the introduction. Vineyards in semi-arid regions or mediterranean climates with a long dry season may not be suitable for moss restoration. Could add this information in the sentences starting around line 67.”* | We will mention the challenges of applying mosses in vineyards already in the introduction. |
| *“Grape yield is also identified as a major risk, but not addressed in this study. If you are able to comment on this it would be interesting, but* | Unfortunately, we have no information about the grape yield at our study site and whether it is affected by different |

| | |
|---|---|
| *otherwise it should be identified as an area for future work."* | management practices. However, when revising the manuscript, we will emphasize the addressed knowledge gaps more strongly and better distinguish them from the knowledge gaps that should be taken into account in the future. |
| *"Along with the risks of water use and yield reduction, another risk that is understated in the current manuscript is the lack of knowledge of how to implement moss restoration in vineyards at scale. This should be treated seriously as a barrier to implementation, and as a knowledge gap which the present work partially addresses."* | The large-scale restoration of mosses has hardly been investigated to date, not only in vineyards but in general. We will focus more on this limitation of the method in the discussion. |
| *"The description of the site's geology can be reduced, as the topsoil is the focus of this manuscript"* | As suggested we shortened the description of the site`s geology. |
| *"Line 54 repace "vines positiviely influenences soil fertility, for example, it increases the" with "vines can positively influence fertility by increasing""* | We changed this sentence as follows: "Additionally, the vegetation cover beneath the vines can positively influence soil fertility by increasing the soil organic carbon content". |
| *"Line 74 replace "they do not require" with "moss does not require""* | We changed this sentence as follows: "Unlike cover crops, mosses do not require mowing, thereby reducing maintenance efforts and costs". |
| *"Line 75 add "mosses may thrive in conditions where …""* | We added "may" in this sentence. |
| *"Line 110 "Overgrown with cover crops" is not clear precisely what was done"* | We rephrased this sentence as follows: "The vineyard produces the Lemberger vine variety, and the soil between the vines is continuously covered with cover crops such as *Lolium perenne*, *Trifolium repens*, *Trisetum flavescens*, and *Achillea millefolium*.". |
| *"Line 143 switch to "three rainfall simulations were conducted on April 13..""* | We removed "at three measurement times" in this sentence. |
| *"Line 178 "June" should be changed to "October""* | Thank you for bringing this typo to our attention! We changed this. |
| *"Lines 216-218 verb tense should be switched to all past tense"* | According to your comment we switched to past tense in this paragraph. |
| *"Line 286 remove comma after Especially"* | We removed the comma here. |

---

## Author Response (AR1)

**Response to referee comments to**

**preprint egusphere-2024-2504: "Effects of moss restoration on soil erosion and soil water content in a temperate vineyard"**

We would like to thank the referees for their helpful comments, which clearly improved our text. We have prepared a revised manuscript where we address all points raised by the reviewers, as described below. Additionally, we conducted changes regarding writing, grammar and comprehensibility. All changes are tracked in the marked-up version of the manuscript.

We would like to follow the reviewer's suggestions and modify the title to "Effects of moss restoration on surface runoff and initial soil erosion in a temperate vineyard".

In this point-by-point reply, reviewer comments are given in grey italic letters in the left column, while our responses are formatted in black as standard text in the right column. Line indications refer to the revised manuscript without marked changes.

**Response to Anonymus Referee #1 (RC1):**

Thank you very much for your thoughtful and positive review. We really appreciate the time and effort you put into your valuable and helpful feedback. Please find the answers to your comments in the table below.

| Reviewer comments | Authors responses |
|---|---|
| *"This manuscript proposes moss restoration as a strategy for reducing erosion in vineyards, and presents the results of a plot-scale field experiment implementing the technique. Moss restoration, bare soil, and grass cover crop are compared, and the rates of runoff, erosion, and infiltration are measured using three simulated rainfall events. The paper demonstrates potential for moss to reduce runoff and erosion, and also identifies gaps for future research. Overall, the quality of the writing is very strong, and the content is presented very clearly. The introduction is particularly concise and comprehensive, to the great benefit of the manuscript."* | Thank you for the overall positive feedback! |
| *"While the central findings of runoff and erosion are solid, the manuscript could still be improved with changes to the data presentation and discussion. The most serious shortcoming of this manuscript is the conditions of the field trial itself. The study site previously had mixed grass and moss in the interr-row, not fallow or cultivated soil as in standard practice. Thus the initial soil conditions may have been different than typical vineyard soils. As the manuscript describes, the weather during the beginning of the trial was unusually warm and dry, leading to challenges establishing the moss matts. Combined with the fact that the "moss"* | We are aware of the difficulties to disentangle the effects of the moss itself and the underlying jute fleece and we mentioned this already in the original preprint. However, we discussed this issue in more detail in **lines 298 – 308**.

Regarding your comment on the initial soil conditions in the vineyard, we would like to point out that our rainfall simulation experiments did not take place in the inter-row (between the rows of vines) but in-row (between the vines within a row). In this area of the vineyard, the soil is typically not tilled, but is treated with herbicides to |

| | |
|---|---|
| *treatment was implemented by spreading moss-impregnated burlap over the soil, it is difficult to disentangle the effects of the burlap from the moss, especially in the first two rain simulation events."* | control plant growth so that the soil surface remains undisturbed. We included photos of the vineyard in the revised manuscript to help readers visualize the location of our treatments in the in-rows (**Figure 1** and **Figure 2**). |
| *"The manuscript could be improved by discussion of how widely-applicable moss could be to vineyards, given the range of aridity experienced in wine-growing regions."* | We agree that discussing the applicability of moss restoration to vineyards across diverse wine-growing regions is important, particularly given the variation in aridity. In the revised manuscript, we expanded the discussion to address this point (**lines 93 – 100; lines 262 – 278**). |
| *"The experimental design and results clearly demonstrate the runoff and erosion rates for the various treatments, but the treatment of long-term soil moisture is lacking in the present manuscript. The introduction identifies that concern about water competition is a primary barrier to implementing cover crops, but the data presented in the manuscript does not effectively characterize the water use of the three treatments. Soil moisture data is only presented for 3 points in time, and at 5 millimeter depth. This zone is particularly sensitive to recent weather, and is not necessarily representative of the water availability to roots. The formulation of hypothesis 3 and presentation of 30-minute time series of water content during simulated rain does not capture the important hydrological properties of the treatment."* | Additionally, we have a continuous dataset with air temperature (15 cm above soil surface), soil temperature (2 cm and 6 cm below soil surface), and soil water content (approx. 14 cm below soil surface) captured every 5 minutes for 21 months (described in lines 137-140 of the original preprint). These data are currently being analysed and are too extensive to be included in this manuscript, which is why we decided to publish these data later in a second manuscript.

 Based on your comment, we decided to focus this manuscript on the effects of moss restoration on surface runoff and soil erosion, and also on the measure of using moss mats in vineyards for erosion control itself. We therefore moved the information on the soil water content in a reduced form to the **supplements**. |
| *"Hypotheses 1 and 2 are very strong, but hypothesis 3 seems less strongly motivated. From the data collected for soil moisture at 5mm, perhaps there could be some analysis of infiltration rate at the surface that would be interesting. But I would suggest that hypothesis 3 and the associated data could be moved to the supplement, and the discussion could be relatively unchanged. This could also allow the information in figures 1 and 2 to be streamlined or moved to the supplement."* | As already mentioned before, we excluded hypothesis 3 from our manuscript and show the respective data in the **supplements**. However, we kept Figure 1 (now **Figure 3**) in the manuscript, as the information about the weather conditions before the moss mats were installed is important to understand their development at the study site. |
| *"The introduction very nicely identifies the state of knowledge for moss restoration and erosion. However, I think the readability would be improved by specifically acknowledging knowledge gaps, especially those that will be addressed in this study."* | Thank you for this suggestion. We revised the introduction to explicitly highlight the key knowledge gaps, such as the very limited knowledge of moss applications in agricultural settings like vineyards and their potential to mitigate soil erosion under different soil management practices (**lines 45 – 47; lines 78 – 79; lines 99 – 100**). |

| | |
|---|---|
| *"The introduction rightly identifies water consumption as a major risk of moss/cover crops, but the experiment mostly does not address this topic. I think this would be great to identify as a knowledge gap in the discussion for future work."* | Following your comment, we shortened the information on the risk of water consumption by mosses and cover crops in the introduction and moved it to the discussion, where we go into more detail and highlight this aspect as a knowledge gap that should be addressed in the future (**lines 339 – 345**). |
| *"The applicability of moss to vineyards should be better addressed in the introduction. Vineyards in semi-arid regions or mediterranean climates with a long dry season may not be suitable for moss restoration. Could add this information in the sentences starting around line 67."* | We now mentioned the challenges of applying mosses in vineyards already in the introduction (**lines 93 – 100**). |
| *"Grape yield is also identified as a major risk, but not addressed in this study. If you are able to comment on this it would be interesting, but otherwise it should be identified as an area for future work."* | Unfortunately, we have no information about the grape yield at our study site and whether it is affected by different management practices. However, when revising the manuscript, we emphasized the addressed knowledge gaps more strongly and better distinguished them from the knowledge gaps that should be taken into account in the future (**see revised introduction**). |
| *"Along with the risks of water use and yield reduction, another risk that is understated in the current manuscript is the lack of knowledge of how to implement moss restoration in vineyards at scale. This should be treated seriously as a barrier to implementation, and as a knowledge gap which the present work partially addresses."* | The large-scale restoration of mosses has hardly been investigated to date, not only in vineyards but in general. We focused more on this limitation of the method in the discussion (**lines 262 – 278**). |
| *"The description of the site's geology can be reduced, as the topsoil is the focus of this manuscript"* | As suggested we shortened the description of the site`s geology (**lines 110 – 112**). |
| *"Line 54 repace "vines positiviely influences soil fertility, for example, it increases the" with "vines can positively influence fertility by increasing""* | We changed this sentence as follows: "Additionally, the vegetation cover beneath the vines can positively influence soil fertility by increasing the soil organic carbon content" (**lines 54 – 55**). |
| *"Line 74 replace "they do not require" with "moss does not require""* | We changed this sentence as follows: "For example, unlike cover crops, mosses do not require mowing, thereby reducing maintenance efforts and costs" (**lines 94 – 95**). |
| *"Line 75 add "mosses may thrive in conditions where …""* | We added "may" in this sentence (**lines 95 – 96**). |
| *"Line 110 "Overgrown with cover crops" is not clear precisely what was done"* | We rephrased this sentence as follows: "The vineyard produces the Lemberger vine variety, and the soil between the vines is continuously covered with cover crops such as *Lolium perenne, Trifolium repens, Trisetum* |

| | |
|---|---|
| | *flavescens*, and *Achillea millefolium*." (**lines 109 – 110**). |
| *"Line 143 switch to "three rainfall simulations were conducted on April 13..""* | We removed "at three measurement times" at this point and placed the explanation in parantheses at the end of the sentence (**lines 145 – 146**). |
| *"Line 178 "June" should be changed to "October""* | Thank you for bringing this typo to our attention! Due to the revision of the manuscript, this paragraph has been removed. |
| *"Lines 216-218 verb tense should be switched to all past tense"* | According to your comment we switched to past tense in this paragraph (**lines 202 – 209**). |
| *"Line 286 remove comma after Especially"* | We removed the comma here (**line 252**). |

**Response to Jesús Rodrigo-Comino (RC2):**

Thank you for taking the time to evaluate our work in detail and we are glad about the overall positive assessment. Your constructive comments provide a strong basis for significant improvements to our manuscript. We have answered your comments individually in the table below.

| Reviewer comments | Authors responses |
|---|---|
| *"Thank you for submitting your manuscript for review. I found the paper to be interesting and novel, offering a valuable contribution to the field of vineyard management."* | Thank you very much for the overall positive evaluation! |
| *"1.    Rainfall Simulations: Please consistently refer to "experiments" when discussing rainfall simulations."* | We now consistently refer to "rainfall simulation experiments" in the manuscript (**For example, line 144, 145, 154, ... , 334**). |
| *"2.    Climate: Given the limited timeframe of your data, I recommend avoiding broad generalizations about climate."* | We fully agree that the data period we have chosen is far from sufficient to talk about climate. Therefore, we now consistently refer to "weather conditions" instead (**Figure 3, Figure S1 of the supplement, and throughout the revised manuscript: lines 164, 165, 182, 249, etc.**). |
| *"3.    Vineyard-Specific References: Consider including more references related specifically to vineyards in your discussion, particularly when comparing your results to those of forests. There is a wealth of literature on rainfall simulation experiments with different soil management practices and countries."* | Thank you for your suggestion. We agree that incorporating more references specific to vineyards would strengthen the discussion. To address this, we have already reviewed additional literature on rainfall simulation studies in vineyards across different countries and under various management practices. We incorporated these references into the discussion section, highlighting similarities and differences in sediment discharge, and surface runoff compared to our study (**lines 280 – 297; lines 334 – 345; 353 – 361**). |
| *"4.    Visual Elements: Please include photos of the vineyard, a map, and a detailed soil description (using English terms)."* | We included a localisation map and photos of the vineyard in different seasons (**Figure 1**). Furthermore, we added **Table 1** with the |

| | general soil characteristics of the two soil horizons identified and translated the soil type Rigosol (German classification) to a Mollic Anthrosol (Relocatic) using the IUSS Working Group WRB (2022). |
|---|---|
| *"5.     Rainfall Simulation Frequency: It would be helpful to justify the decision to conduct only three rainfall simulations in a single year. Some readers may question the representativeness of this data. If you used a large-scale rainfall simulator, please provide photos and explain the logistical challenges that may have limited the number of experiments. I perfectly know them, but not all readers must be familiar with this."* | In total, we conducted 36 rainfall simulations at three measurement times in one year. This means each rainfall simulation experiment comprises 12 individual rainfall simulations. The decision to conduct three rainfall simulation experiments in a single year was influenced by practical and logistical constraints, so the scope of the experiments was adapted to the time, equipment and personnel available. Our scientific idea behind this decision was to measure the influence of vine foliage on soil erosion. However, as the moss mats had not yet fully established themselves in the vineyard in June, as we had originally expected, we carried out an additional rainfall simulation experiment in October. To provide readers with more clarity, we included photos of the rainfall simulator (**Figure 2**) and a more detailed explanation of the rainfall simulation experiments (**lines 146 – 148**). |
| ***Specific comments from the provided PDF*** | |
| *Line 1 (Title): "No clear what you mean when you use rainfall simulation experiments*

*Also, why in temperate climate is necessary water retention? (I know now the current conditions, but in the abstract and goals this is missed)"* | Due to the revision of the manuscript, the hypothesis, results, and discussion regarding our near-surface soil water content measurements were moved to the supplement, , which is why we don`t go into detail in the abstract. |
| *Line 1 (Title): "Rainfall simulation does not show soil erosion, they show activation of…"* | We changed our title to "**Effects of moss restoration on surface runoff and initial soil erosion in a temperate vineyard**" and also considered this comment within the manuscript, especially in the methods section (**lines 145 - 149**). |
| *Line 10: "Specify this"* | We have decided not to use the term "fragile soils" in the abstract, as we don`t want to provide a detailed explanation at this point (a detailed explanation is given in the introduction: **lines 37 - 39**). Instead we rephrased and shortened the initial paragraph of the abstract and focused more on the description of the methodologies applied (**lines 9 -17**). |
| *Line 12: "Can be"* | We rephrased this sentence as follows: "Therefore, the search for alternative management practices becomes vital, and |

| | vegetation covers, including mosses, have the potential to reduce soil erosion." (**lines 11 - 12**). |
|---|---|
| *Line 13: "In other crops is usual?"* | The use of other crops as an erosion control measure has certainly not been conclusively studied and there is still a great need for further research. However, there are far fewer studies regarding moss restoration as erosion control, especially in viticulture. We highlighted this by rephrasing this sentence as follows: "However, research on moss restoration as erosion control is still in its infancy, and has never been applied in vineyards." (**lines 12 - 13**). |
| *Line 16: "I would introduce reducing other parts, which kinds of rainfall simulation experiments you did"* | According to your recommendation, we shortened the initial paragraph of the abstract and added more information on the rainfall simulation experiments (**lines 15 – 17**). |
| *Line 40: "Possibly, I would use another type of review or European comparison instead of this paper of my colleague Marcella, which is a study case, try to find something more representative"* | We removed this reference at this point and added the following two references instead (**line 39**): (Prosdocimi et al., 2016; Rodrigo Comino et al., 2016) |
| *Line 43: "This author has other papers to be cited with bare soils, try to diversify the references"* | We removed this reference at this point and included the following two references instead as they show the significant impact of bare soils on soil erosion in vineyards (**line 42**): (Rodrigo-Comino et al., 2018; Rodrigo Comino et al., 2015) |
| *Line 132: "It would be great to include photos"* | We provided a series of photos of the moss mats from their application in the field to the last rainfall simulator experiment in the supplement (**Figure S2**). In this way, the development of each individual replicate of the moss mats can be followed very well over time. |
| *Figure 2: "It would be interesting to see what happen in the middle of these months, now, you are not showing if between months you had extreme changes"* | We provided a diagram for precipitation and temperature of the entire measurement period from February to October 2022 in the supplement (**Figure S1**). |
| *Line 373: "I do not know if they are comparable, in Madrid, the climate conditions are really arid"* | During the review process we decided to focus this paper more on sediment discharge and surface runoff and moved the part about soil water content to the supplement. Here, we have kept the reference Marques et al. (2020), as we also point out in the lines **91-92** of the supplements that the different responses of the soil water content could also be due to different climatic conditions of the two the two research sites (Spain & Germany). |

IUSS Working Group WRB (Ed.) World Reference Base for Soil Resources. International soil classification system for naming soils and creating legends for soil maps, International Union of Soil Sciences, Vienna, Austria, https://wrb.isric.org/files/WRB_fourth_edition_2022-12-18_errata_correction_2024-09-24.pdf (last access: 19.11.2024), 2022.

Marques, M., Ruiz-Colmenero, M., Bienes, R., García-Díaz, A., and Sastre, B.: Effects of a permanent soil cover on water dynamics and wine characteristics in a steep vineyard in the Central Spain, Air, Soil and Water Research, 13, 1-10, https://doi.org/10.1177/1178622120948069, 2020.

Prosdocimi, M., Cerdà, A., and Tarolli, P.: Soil water erosion on Mediterranean vineyards: A review, Catena, 141, 1-21, https://doi.org/10.1016/j.catena.2016.02.010, 2016.

Rodrigo-Comino, J., Novara, A., Gyasi-Agyei, Y., Terol, E., and Cerdà, A.: Effects of parent material on soil erosion within Mediterranean new vineyard plantations, Engineering Geology, 246, 255-261, https://doi.org/10.1016/j.enggeo.2018.10.006, 2018.

Rodrigo Comino, J., Brings, C., Lassu, T., Iserloh, T., Senciales, J. M., Martínez Murillo, J. F., Ruiz Sinoga, J. D., Seeger, M., and Ries, J. B.: Rainfall and human activity impacts on soil losses and rill erosion in vineyards (Ruwer Valley, Germany), Solid Earth, 6, 823-837, https://doi.org/10.5194/se-6-823-2015, 2015.

Rodrigo Comino, J., Iserloh, T., Lassu, T., Cerdà, A., Keestra, S. D., Prosdocimi, M., Brings, C., Marzen, M., Ramos, M. C., Senciales, J. M., Ruiz Sinoga, J. D., Seeger, M., and Ries, J. B.: Quantitative comparison of initial soil erosion processes and runoff generation in Spanish and German vineyards, Science of The Total Environment, 565, 1165-1174, https://doi.org/10.1016/j.scitotenv.2016.05.163, 2016.